# Improved Self-Explanatory Graph Learning Method Based on Controlled Information Compression and Branch Optimization

## Abstract

Graph Neural Networks have gained widespread application across various domains and have motivated research into their explainability. Self-explainable methods consider inherent explanations during prediction and provide insights to reveal the decision-making processes. However, the transparent explainability of these methods often comes at the cost of predictive performance. One reason is that these methods suffer from a distribution shift when directly using explanation subgraphs to make predictions. In this work, we propose **S**elf-expl**A**inable **G**raph l**E**arning (SAGE) to improve the performance of self-explainable methods. Specifically, SAGE learns attention weights for edges to guide message passing, generating more meaningful and discriminative representations. In this process, we emphasize label-relevant critical structures while diminishing the influence of noisy ones. Additionally, we control the degree of noisy information compression applied to the subgraphs by establishing a lower bound for the attention scores of irrelevant noisy structures, which helps reduce the deviation from the original graph and mitigates the distribution shift. Furthermore, we introduced an optional strategy called branch optimization, exploring the optimal GNN state to improve the model's optimization effectiveness. Experimental results on real-world datasets demonstrate that SAGE can achieve predictive accuracy comparable to or even higher than baselines. Compared to the backbone model, our self-explainable framework attains an average performance improvement of 10.5% across four datasets.

## Introduction

Graph Neural Networks (GNNs) have emerged as powerful tools for representation learning of graph-structured data, such as social networks (Dwivedi et al. 2023), citation networks (Kipf and Welling 2016), and chemical molecular graphs (Jumper et al. 2021). However, their expressivity is often built upon a highly nonlinear entanglement of irregular graph features. This leads to the black-box nature of GNNs that limits their application in critical tasks requiring trust. There have been numerous works proposed for explaining trained GNNs by extracting salient substructures of input graphs as explanations in a post-hoc manner (Luo et al. 2020; Ying et al. 2019). Specifically, these methods work on a pre-trained GNN model and propose different types of combinatorial search methods to explore the subgraphs of the input data critical to the GNNs' predictions. Although these methods can provide relatively accurate explanations, they are recognized to be biased or inconsistent in revealing the actual reasoning process of the original model (Dai and Wang 2021). In contrast to post-hoc methods, self-explainable methods can provide explanations while making predictions, which ensures the derived explanations are inherently consistent with the trained model's architecture. However, the built-in transparent explanations often come at the cost of prediction accuracy (Du, Liu, and Hu 2018). One reason is the distribution shift problem when evaluating or using the explanation subgraphs. Actually, both types of methods mentioned above suffer from distribution shift. For instance, treating the prediction on the subgraph as the subgraph's label could potentially affect the model's performance since it may significantly deviate from the actual ground truth. In other words, directly using explanatory subgraphs for prediction often fails to achieve ideal results. Due to the significant difference in data distribution between the explanatory subgraphs and the original graphs, there exists a distribution shift. Therefore, mere explanatory subgraphs are not necessarily the optimal graph structures for prediction. From this perspective, we believe that the noise structures in the graph should not be completely eliminated, but rather mitigated to a certain extent, as these components, while not essential, are useful for prediction.

In this work, we propose SAGE to address this issue. SAGE controls the degree of distribution shift by properly compressing noisy information. Specifically, it learns attention weights for each edge while driving the data distribution of these attention weights to gradually approach an ideal distribution we set, in which each edge has a certain sampling probability, such as 70%. This operation sets a boundary control for attention weights of all edges. During the optimization process, the model gradually allocates higher attention weights to critical structures relevant to the label, thereby achieving better classification performance. Through the above boundary control, we reduce the negative impact of distribution shift as the compressed subgraph retains at least 70% of the information from the original graph. With the above configuration, SAGE completes the subgraph selection. In the subsequent subgraph learning process, unlike previous methods that directly combine the attention weight matrix with the original graph, we ap-

ply attention to the message-passing process at each layer of the GNN, further emphasizing the information of critical structures. We evaluated the proposed SAGE framework on graph classification tasks. Experimental results show that SAGE demonstrates superior subgraph discovery capabilities while achieving classification performance comparable to or even better than baselines. We summarize our contributions as follows:

- We control the degree of compression applied to noisy information by setting lower bounds on the attention weights for edges. This approach can mitigate the distribution shift problem and the learned attention will be used to guide neighborhood aggregation for final embeddings.

- The unified self-explainable GNN framework can make predictions and provide edge-level explanations in an end-to-end manner. By learning proper attention weights, we can obtain more meaningful embeddings which can effectively enhance the performance of the backbone model.

- To further enhance the performance based on the framework, we introduce an optional optimization strategy called Branch optimization. This strategy explores better GNN states in a branch-like manner without affecting the original optimization process.

- Compared with classic GNNs and self-explainable methods, experimental results on real-world datasets demonstrate the effectiveness of our framework in graph classification tasks. The visualization results also demonstrate that SAGE can provide accurate explanations.

## Related Work

Some self-explainable methods employ the Graph Information Bottleneck(GIB) principle to identify informative subgraphs for explanation. These methods impose constraints on the information of subgraphs. IB-subgraph (Yu et al. 2020) proposes a GIB objective and adopts a bi-level optimization strategy. Specifically, in the inner optimization loop, the IB-subgraph first utilizes a GNN to obtain representations of both the original graph and the subgraph. It then trains a parameter network to estimate the mutual information between the original graph and the subgraph at the current epoch. Subsequently, IB-subgraph optimizes the subgraph to minimize the mutual information in the outer optimization loop. Since this bi-level optimization process is inefficient and unstable, another method VGIB (Yu, Cao, and He 2022) proposes a new framework. It reformulates the subgraph identification problem into two steps: graph perturbation and subgraph selection. Noise is injected into node representations according to a learned probability to suppress information in the original graph. If informative subgraphs are injected with noise, the classification loss will increase. The VGIB objective is given a tractable variational upper bound, resulting in superior empirical performance and theoretical properties. Consequently, driven by the loss function, VGIB further obtains the desired subgraph by filtering out noise from the perturbed graph. As VGIB only

learns node-level scores and directly reads out graph embeddings, which may adversely affect the quality of graph representations. GSAT (Miao, Liu, and Li 2022) injects randomness when learning attention to disrupt spurious correlations in the data. It samples random attention from a Bernoulli distribution based on logits computed by an MLP. GSAT uses this random attention to extract explanatory subgraphs and make predictions. It also incorporates a reasonable regularization term to control randomness, which is equivalent to controlling information from an information theory perspective. This random attention mechanism endows GSAT with excellent robustness and generalization capabilities.

## Preliminaries

### Graph Neural Network

We denote a graph by $G = (V, E)$ where $V$ is the node set and $E$ is the edge set. Let $X \in R^{|V| \times F}$ denote the node feature matrix where $F$ is the input feature dimension of nodes and the $i$-th row of X is the feature vector $x_i$ of node $v_i$. The graph structure can be described by an adjacency matrix $A \in \{0, 1\}^{|V| \times |V|}$, where $A[i, j] = 1$ if edge $(v_i, v_j)$ exists, otherwise $A[i, j] = 0$.

A GNN takes a graph $G$ as input and learns node representations which can be used in downstream tasks. Typically, initial node representations are node feature matrix $X$ and GNN updates them by employing the message-passing mechanism to propagate and aggregate information from neighbors. For an $L$-layer GNN, the representation of a target node is jointly determined by its $L$-hop neighbors. Different GNNs may adopt various aggregation and updation strategies. Formally speaking, for a given GNN, the hidden state $h_i^{(l)}$ of node $v_i$ at layer $l$ can be represented as:

$$h_i^{(l)} = UPDATE^{(l)} \left\{ h_i^{(l-1)}, AGG^{(l)} \left( \left\{ h_j^{(l-1)}, \forall v_j \in \mathcal{N}_i \right\} \right) \right\},$$

where $\mathcal{N}_i$ is the set of neighbors of node $v_i$. $UPDATE^{(l)}(\cdot)$ is an update function such as $ADD$ or a linear transformation after concatenation, while $AGG^{(l)}(\cdot)$ is a function aggregating representations of neighbor nodes, such as taking the average or using a multi-head attention mechanism.

In this work, we focus on the graph classification task. The READOUT function aggregates node representations from the final layer to obtain the entire graph's representation.

$$h_G = READOUT \left( \{ h_v \mid v \in V \} \right),$$

where the READOUT function can be a simple summation or other more sophisticated graph-level pooling operations. Given a set of input graphs $\{G_1, ..., G_N\}$ and their labels $\{y_1, ..., y_N\}$, graph classification aims to learn a representation vector $h_G$ to predict the label of the entire graph, $\hat{y}_i = f(h_{G_i})$.

### Problem Formulation

We follow previous works(Miao, Liu, and Li 2022; Sui et al. 2022) and provide attention scores for edges as edge-level explanations. This paper aims to develop a self-explainable GNN framework including a GNN encoder $f_\theta$, an explanation extractor $g_\phi$. Given a graph $G$ and node features $X$,

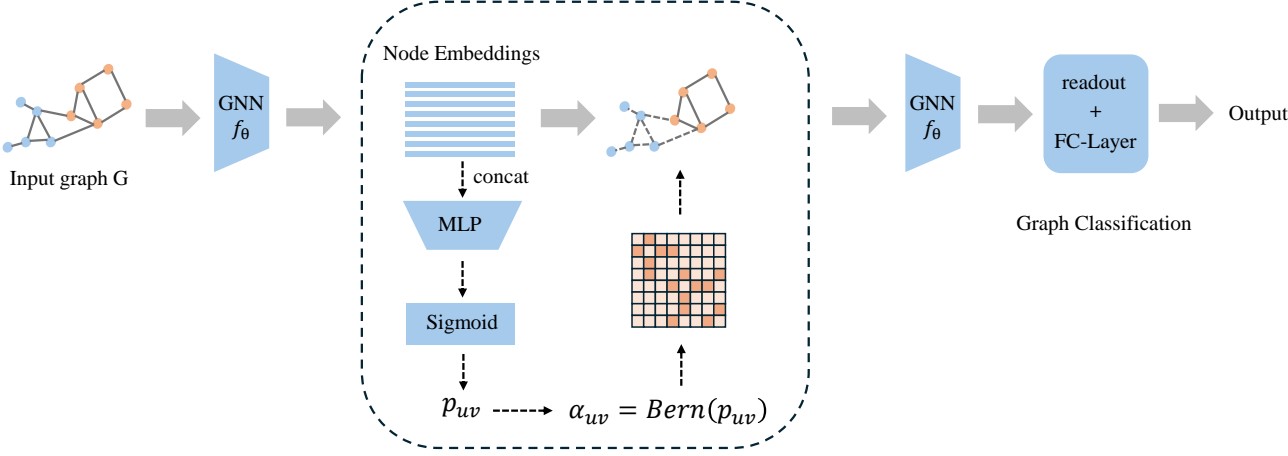

Figure 1: Illustration of the proposed SAGE framework. $f_\theta$ encodes the edges of the input graph. Through MLP and Gumbel-softmax, it obtains the attention score distribution for edges in order to guide the aggregation.

the framework is able to provide faithful explanations using $G_s = g_\phi(G, X)$ and accurate predictions with $\hat{y} = Classify(f_\theta(G_s, X))$.

## Methodology

### Framework Overview

As shown in Fig.1, we adopt a simple framework including a GNN encoder $f_\theta$ and a feature extractor $g_\phi$. Firstly, the $f_\theta$ is pre-trained and encodes the input graph $G$ into a set of node representations $\{h_v | v \in V\}$. For each edge $(u, v)$, we consider the concatenation $[h_u, h_v]$ as its representation. Then the extractor $g_\phi$, an MLP layer plus sigmoid, maps the concatenation into $p_{uv} \in [0, 1]$, which can be considered as a form of attention score.

$$p_{uv} = Sigmoid\left(Extractor\ g_\phi(\text{concat}[h_u, h_v])\right), \quad (1)$$

The sampling operation based on $p_{uv}$ is not differentiable. In our implementation, we use the Gumbel-softmax trick to reparameterize this step.

$$\alpha_{uv} \sim Bern\left(p_{uv}\right), \quad (2)$$

With $\alpha_{uv}$ for all the edges, we obtain $\alpha$ representing the learned attention weights and we apply $\alpha$ to each layer of the $f_\theta$ to guide the message-passing process. Finally, we obtain a more meaningful graph representation and make predictions.

$$\hat{y} = \text{Classify}\left(GNNf_\theta\left(A, X, \alpha\right)\right). \quad (3)$$

### Branch optimization

Typically, the extractor module and the GNN encoder are updated simultaneously. However, this approach may not allow the GNN to fully adapt to the changes in the extractor, potentially limiting the model's performance. Based on this intuition, we propose a training strategy called Branch optimization to further enhance the performance of the self-explainable pipeline mentioned above. During the training

process, we save the current checkpoint after backpropagation and parameter updates. Then, we freeze the parameters of the extractor $g_\phi$ and continue to optimize the GNN $f_\theta$'s parameters several times according to the hyperparameter 'inner-epoch'. During this optimization process, if better GNN states are obtained, they are recorded and saved. After this, we load the checkpoint and restore the original state, ensuring that the optimization process in each epoch functions like a branch, without affecting the original training process. Considering the increased computational cost associated with the additional optimization process, we perform branch optimization every few epochs or when the saved best metric updates. The entire process is illustrated in Algorithm 1.

---

**Algorithm 1:** branch optimization

**Input:** $GNNf_\theta, G = (V, E, X, Y)$.
1 Save checkpoint.;
2 Fix parameters of extractor $g_\phi$.;
3 **for** *each inner-epoch* **do**
4     $Z \leftarrow GNNf_\theta(G, X)$;
5     Calculate attention weights matrix $\alpha$.;
6     $\hat{Y} \leftarrow GNNf_\theta(G, X, \alpha)$;
7     Compute loss and update parameters normally. ;
8     Test the model and record the results.;
9 **end**
10 Load checkpoint.;

---

### Information compression mechanism

Firstly, we choose the GIN as our backbone model to obtain high-quality node embeddings. Since the learned attention scores are nondifferentiable, we employed the commonly used Gumbel-softmax as a reparameterization trick which also introduces randomness to the model and enhances its

**Algorithm 2:** Training SAGE for graph classification

---

**Input:** The input graph $G = (V, E)$, node features $X$, node labels $Y$, GNN encoder $GNN f_\theta$, feature extractor $g_\phi$.

**1 for** *each epoch* **do**

**2**      $Z \leftarrow GNN f_\theta(G, X)$.

**3**      Obtain edge embeddings via concatenating.

**4**      $\alpha \leftarrow$ Calculate attention weights matrix.

**5**      $\hat{Y} \leftarrow GNN f_\theta(G, X, \alpha)$.
        `// α works as attention in each layer.`

**6**      Compute loss with Eq.4.

**7**      Update parameters with backpropagation.

**8**      Start branch optimization every few epochs.

**9 end**

**Output:** Prediction Y, edge importance $\alpha$ for explanation.

---

robustness. Unlike previous methods that directly combine the attention weight matrix $\alpha$ and the original graph $G$, our approach, we applied the attention scores to each layer of the GNN to guide the message aggregation of neighbor nodes. This process further differentiates the contributions of key structures and the remaining parts to the final graph embedding. The overall pipeline of SAGE is depicted in Algorithm2.

Our loss function consists of two components: the first component is the cross-entropy representing the classification loss, and the second component is the Kullback-Leibler (KL) divergence between the learned attention score distribution and a predefined ideal distribution:

$$\min_{\theta, \phi} -\mathbb{E}\left[\log \mathbb{P}_\theta(Y|G_S)\right] + \beta \mathbb{E}\left[\mathrm{KL}(\mathbb{P}_\phi(G_S|G)||\mathbb{Q}(G_S))\right], \tag{4}$$

where $\mathbb{Q}(G_S)$ is a Bernoulli distribution in which the sampling probability of each edge approaches $r$ and $r \in [0, 1]$ is a hyperparameter. The overall pipeline of SAGE is depicted in Algorithm2.

As shown in Fig.2, the KL divergence component encourages the learned attention weights for each edge to approach r, such as 0.7, while the cross-entropy component drives the weights of structures critical to prediction towards 1. Driven by the loss function, the attention weights of structures irrelevant to the prediction are compressed to $r$, and the contributions of critical structures are further highlighted through a message-passing process. The model optimizes the allocation of attention scores and gradually improves classification accuracy. As we only reduce the weights of task-irrelevant structures, this approach does not significantly alter the data distribution of the original graph, thereby controlling the impact of distribution shift on the model. Through this method, we compressed task-irrelevant noise information.

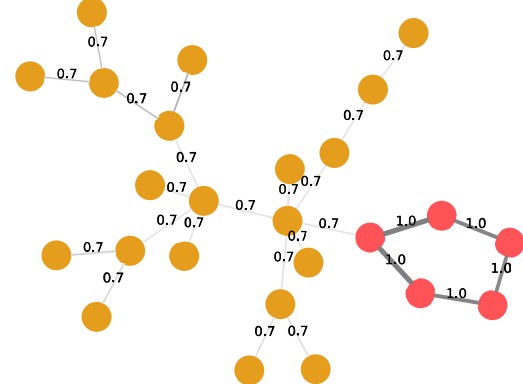

Figure 2: Illustration of the expected compression results of the objective function

Table 1: Statistics of datasets.

| Dataset | Graphs | Nodes | Edges | Classes |
|---|---|---|---|---|
| MUTAG | 188 | 17.93 | 19.79 | 2 |
| NCI1 | 4110 | 29.87 | 92.3 | 2 |
| PROTEINS | 1113 | 39.06 | 72.82 | 2 |
| IMDB-BINARY | 1000 | 19.77 | 96.53 | 2 |
| MNIST | 70000 | 70.57 | 564.66 | 10 |
| BA-2motifs | 1000 | 25 | 51.39 | 2 |

## Experiments

### Datasets and Baselines

We select real-world datasets such as TUDataset (Morris et al. 2020) and Superpixel graphs (Knyazev, Taylor, and Amer 2019) to evaluate the prediction performance of our model. For TUDataset, We use three biological datasets (MUTAG, NCI1, PROTEINS) and a social networks dataset (IMDB-B), which are commonly used in graph classification tasks. For explainability performance, we select BA-2motifs and MUTAG since they provide ground truth motif. Their statistics are presented in Table.1.

For datasets from the TUDataset, we adopt a random split of 80% for training and 10% each for validation and testing. In Mutag, we use mutagen molecules with -NO2 or -NH2 to evaluate or visualize explanations (because only these samples have explanation labels). For MNIST-75sp, we use the default splits following (Knyazev et al., 2019); due to its large scale in the graph setting, we also reduce the number of training samples to speed up training.

We compare prediction performance with three classic GNN models GCN(Kipf and Welling 2016), GAT(Veličković et al. 2018), and GIN(Xu et al. 2019), and three inherently interpretable models IB-subgraph(Yu et al. 2020), VIB-GSL(Sun et al. 2021), and GSAT(Miao, Liu, and Li 2022).

### Setup

Here, we introduce the detailed experimental setup: **(1) Batch size.** For MNIST-75sp, we use a batch size of 256

| model | year | MUTAG | NCI1 | PROTEINS | MNIST | IMDB-B |
|---|---|---|---|---|---|---|
| GCN | 2016 | 74.50±7.9 | 73.16±3.5 | 72.83±4.2 | 90.49 | 70.70±3.7 |
| GIN | 2019 | 80.50±7.9 | 75.04±2.1 | 70.30±4.8 | 95.74±0.4 | 73.20±4.8 |
| GAT | 2018 | 73.50±7.4 | 66.05±1.0 | 71.35±4.8 | 95.53 | 71.30±4.2 |
| IB-subgraph | 2020 | 83.90±6.4 | 64.65±6.8 | **74.90±5.1** | 93.10±1.3 | 73.70±7.0 |
| VIB-GSL | 2021 | 81.00±6.6 | 63.75±3.4 | 73.66±3.3 | - | **77.10±1.4** |
| GSAT | 2022 | 95.80±4.2 | 68.13±2.6 | 71.10±3.7 | 96.24 ± 0.2 | 75.40±2.9 |
| SAGE (Ours) | | **97.10±1.5** | **78.50±2.1** | 71.60±3.0 | **96.50±0.1** | 77.00±3.3 |
| Average impro (%) | | 20.0 ↑ | 15.2 ↑ | - | 2.5 ↑ | 4.8 ↑ |

Table 2: Graph Classification Performance. We report the mean and standard deviation of the testing accuracy. The bold results are the overall best performances and the underlining results are the second-best performances. Some data come from (Sui et al. 2022; Seo, Kim, and Park 2024).

to speed up training due to its large scale; all other datasets use a batch size of 128. **(2) Epoch.** Except for MNIST-75sp training for 200 epochs, all other datasets are trained for 100 epochs and we report the metric of the epoch that achieves the best validation performance. During the pre-train process, we employ the same strategy of using the models that achieve such best performance on the validation set as the pre-trained model. **(3) Learning Rates.** We employ the Adam optimizer configured with a learning rate of 1e-3 to update the parameters of the network, including GIN and MLP. In branch optimization, We use another Adam optimizer with a learning rate of 5e-4 to update the parameters of GIN temporarily. **(4) Backbone.** We use a two-layer GIN with 64 hidden dimensions and a 0.3 dropout ratio to encode the input graph. For graph classification, we employ a global-add-pool from Pytorch to obtain the embedding of the entire graph. **(5) Hyperparameters.** We set the inner-epoch to 3 since we only intend to slightly adjust the parameters of the GIN; We set the $\beta$ and r to 1, 0.7 for all datasets respectively.

To accelerate training, the classification results on MNIST-75sp are averaged over 5 runs and all other experiment results are averaged over 10 times tests with different random seeds.

## Graph Classification

To evaluate the effectiveness of the SAGE on graph classification, we compare our model with classic GNNs and self-explainable models. We report the classification accuracy in Table 2. As shown in Table 2, SAGE outperforms baselines on the MUTAG, NCI1, and MNIST-75SP, while achieving suboptimal results on the IMDB-B. Our model also achieved comparable results on the PROTEINS. Notably, while most explainable baseline models perform poorly on NCI1, our model, in contrast, achieves an average improvement of 12%. The experimental results indicate that SAGE can achieve better predictive performance by compressing the noise information from the original graph.

SAGE learns edge attention weights to guide the message-passing process to obtain more meaningful representations while providing key substructures as explanations. Our framework can enhance the performance of the backbone

model GIN in all the datasets, which means that the attention learned by SAGE is effective.

## Explanation Visualization

Although our primary goal is to achieve better classification by compressing noisy graph information, we can still provide edge attention scores as explanations. To verify the quality of the explanation, we choose two instances from each dataset and present the visualization of explanations in Fig.3. We determine the thickness of edges based on their attention scores. Considering that in our model, the scores of important substructures do not differ greatly from the remaining parts, we apply a normalization to amplify the difference between the scores of the two parts. Naturally, the edges within the motifs identified by the model as most relevant to the prediction are represented by thick black lines. As shown in the figure, SAGE can correctly identify the house-structured and five-node cycle in BA-2motifs, as well as the $NO_2$ and $NH_2$ groups in MUTAG. Notably, there are multiple motifs in the instances from the MUTAG, SAGE can identify almost all of them.

This result aligns with the design of SAGE: 1) critical structures get high attention scores close to 1, allowing them to be effectively retained and highlighted in the compressed graph representation. 2) Noisy structures have their attention scores pushed towards the lower bound r, which compresses and downgrades their influence without completely removing them. By differentiating between key and noisy components in this manner during the prediction process, SAGE can generate compressed graph representations that preserve the most salient information for the downstream task while mitigating noise and irrelevant components. The hyperparameter r provides control over how aggressively noise is suppressed.

## Ablation Studies

In this section, we conduct ablation studies from three perspectives. First, the effect of branch optimization where we obtain the variant "w/o opt" by removing branch optimization. Second, the importance of edge-level explanations where we learn attention scores for nodes and lift node-level attention to edge-level to adapt the following aggregation

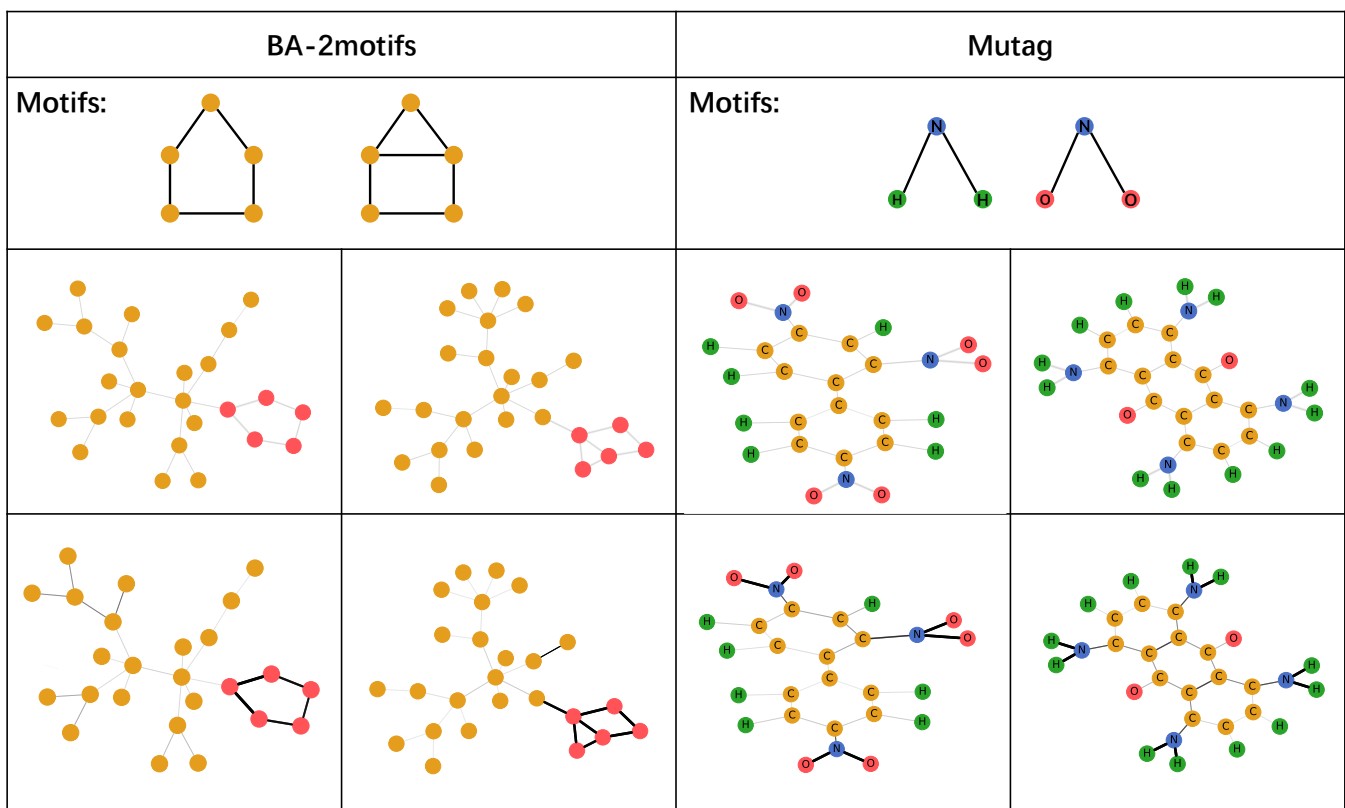

Figure 3: Visualizing graphs compressed by SAGE for Ba-2Motifs and Mutag. Edges colored black are critical structures.

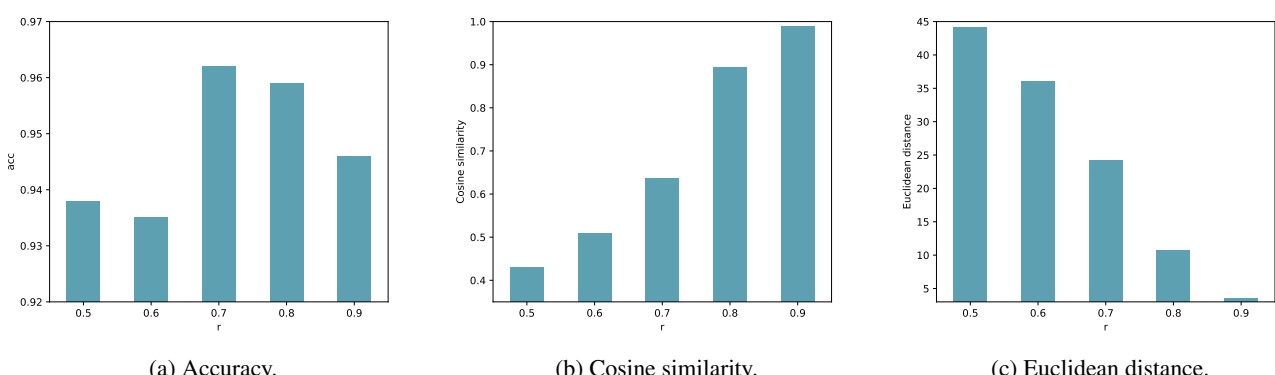

(a) Accuracy.

(b) Cosine similarity.

(c) Euclidean distance.

Figure 4: The impact of the hyperparameter r on the final graph embedding. (a) The effect of r on classification accuracy. (b)(c) represent the cosine similarity and Euclidean distance between the final graph embedding and the original graph embedding, respectively.

process. This variant is named "w/o edge". Third, we evaluate the effect of attention-guided aggregation. In this case, we obtain the compact graph embedding by a weighted sum of all node embeddings based on node-level attention scores. This variant is named "w/o agg".

We test the classification accuracy of SAGE and its three variants on the MUTAG and NCI1. As shown in Table 3, The performance of all three variants has decreased compared to the complete model. When branch optimization is not em-

Table 3: Ablation studies on SAGE.

|  | MUTAG | NCI1 |
|---|---|---|
| w/o opt | 95.80±2.0 | 76.6±1.4 |
| w/o edge | 95.80±1.3 | 74.3±1.5 |
| w/o agg | 93.40±2.8 | 77.7±1.9 |
| with all | 97.10±1.5 | 78.50±2.1 |

ployed, performance declines on both datasets, which validates the effectiveness of branch optimization. The results of the "w/o edge" variant indicate that encoding edge features generates more meaningful attention weights compared with node-level explanation. Furthermore, the performance of the "w/o agg" variant should significantly decrease compared to the complete model which is confirmed by its results on MUTAG. However, the performance drop on NCI1 is minimal, which we attribute to the specific characteristics of this dataset. Each step of our model is designed to optimize the compression of the noise information of the input graph. This ensures that every step contributes to the model's performance, which can be verified from the results of our ablation studies.

The hyperparameter r controls the threshold for attention scores assigned to label-irrelevant structures. It regulates the degree of compression for noisy information and, consequently, the extent of distribution shift between the compressed graph structure and the original graph. To validate the effect of r, we tested the model's performance with r values in the set 0.5, 0.6, 0.7, 0.8, 0.9. The evaluation metrics used are classification accuracy, cosine similarity, and Euclidean distance. The cosine similarity and Euclidean distance between the final graph embedding and the original graph embedding serve as approximations of the similarities between graph distributions. The results are presented in Fig.4. The results show that as the value of r increases, the cosine similarity increases while the Euclidean distance decreases. This indicates that the value of r can, to some extent, control the data distribution of the compressed graph, thereby further reducing the negative impact of the distribution shift. Additionally, the classification accuracy reaches its peak when r is 0.7. This can be interpreted as follows: when r is relatively high, the compression of noisy information is insufficient; conversely, when r is relatively low, the degree of distribution shift is more significant, which also impairs classification performance.

Table 4: The influence of hyperparameter $\beta$ on graph embeddings

|  | Accuracy | Cosine similarity ↑ | Euclidean distance ↓ |
|---|---|---|---|
| $\beta = 0$ | 95.1 | 0.333 | 57.207 |
| $\beta = 0.5$ | 96.0 | 0.630 | 26.276 |
| $\beta = 1$ | **96.2** | **0.637** | **24.224** |

On the other hand, we also conducted a study on the hyperparameter $\beta$ in Eq.4 to evaluate the effectiveness of the second term in Eq.4 that controls the distribution boundary. The results are shown in Table.4 . When $\beta$=0, there are significant changes in cosine similarity and Euclidean distance, indicating that using only cross-entropy classification as the objective function may impair model performance due to significant distribution shifts. When $\beta$=0.5 or 1, all metrics of the model improved, demonstrating the effectiveness of controlling the distribution boundary.

## Conlusion

In this paper, we attempt to improve the performance of self-explainable graph learning by setting boundaries for the learned attention scores of edges. We propose SAGE learning an attention score for each edge in the graph and utilize these scores to guide the message-passing process. Driven by the loss function, the attention scores of all edges are pushed towards the predefined lower bound, while the scores of critical structures gradually approach 1, thereby weakening label-irrelevant structures and emphasizing crucial ones. Additionally, we introduce an optional branch optimization strategy that can further enhance the optimization quality of SAGE to a certain extent. Extensive experiments on real-world datasets demonstrate that SAGE alleviates the distribution shift problem and improves the performance of GNNs. In the future, we aim to extend the model's applicability to more complex graph learning scenarios.

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

# Appendix

## Datasets

We select real-world datasets such as TUDataset (Morris et al. 2020) and Superpixel graphs (Knyazev, Taylor, and Amer 2019) to evaluate prediction performance of our model. For TUDataset, We use three biological datasets (MUTAG, NCI1, PROTEINS) and a social networks dataset (IMDB-B), which are commonly used in graph classification tasks. For explainability performance, we select BA-2motifs and MUTAG since they provide ground truth motif.

- Based on Barabasi-Albert (BA) model, Each base graph in **BA-2Motifs** is attached with a house-like motif or a five-node cycle motif. House motifs and cycle motifs correspond to class labels and thus are regarded as ground-truth explanations for the two classes respectively.

- **MUTAG** is a molecular property prediction dataset, where nodes are atoms and edges are chemical bonds. Each graph represents a chemical compound and is associated with a binary label based on its mutagenic effect. Following (Luo et al., 2020), -NO2 and -NH2 in mutagen graphs are labeled as ground-truth explanations.

- Similar to MUTAG, **NCI1** is relative to anti-cancer screens where the chemicals are assessed as positive or negative for cell lung cancer.

- **PROTEINS** includes proteins that are classified as enzymes or non-enzymes. Nodes represent the amino acids and two nodes are connected by an edge if they are less than 6 Angstroms apart.

- **IMDB-BINARY** derives from movie collaboration. In each graph, nodes represent actors/actresses, and there is an edge between them if they appear in the same movie. These graphs are derived from the Action and Romance genres and each graph is associated with a binary label based on its type.

- **MNIST-75sp** is an image classification dataset, where each image in MNIST is converted to a superpixel graph. Each node in the graph represents a superpixel and edges are formed based on spatial distance between superpixel centers. Node features are the coordinates of their centers of masses. Nodes with nonzero pixel values provide ground-truth explanations. Note that the subgraphs that provide explanations are of different sizes in this dataset.

For datasets from the TUDataset, we adopt a random split of 80% for training and 10% each for validation and testing. In Mutag, we use mutagen molecules with -NO2 or -NH2 to evaluate or visualize explanations (because only these samples have explanation labels). For MNIST-75sp, we use the default splits following (Knyazev et al., 2019); due to its large scale in the graph setting, we also reduce the number of training samples to speed up training.

## Baselines

We compare prediction performance with three classic GNN models GCN, GAT, and GIN, and three inherently interpretable models IB-subgraph, VIB-GSL, and GSAT.

- **GCN** (Kipf and Welling 2016) captures local structural information and update embeddings by aggregating neighboring node features; **GAT** (Veličković et al. 2018) introduces an attention mechanism to assign different weights to neighbors during aggregation; **GIN** (Xu et al. 2019) is probably the most expressive among the class of GNNs and is as powerful as the Weisfeiler-Lehman graph isomorphism test.

- **IB-subgraph** (Yu et al. 2020) derives an optimizable objective from a mutual information estimator for irregular graph data and proposes the graph information bottleneck (GIB) framework for discovering a critical substructure.

- **VIB-GSL** (Sun et al. 2021) advances the Information Bottleneck (IB) principle for graph structure learning and proposes a new variational IB-guided framework which jointly optimizes the graph structure and graph representations.

- **GSAT** (Miao, Liu, and Li 2022) injects stochasticity to edges and leverages the reduction of stochasticity to select important edges under the guidance of the information bottleneck principle.