# OpenReview forum: "Improved Self-Explanatory Graph Learning Method Based on Controlled Information Compression and Branch Optimization"
_AAAI.org/2025/Workshop/NeurMAD — AAAI 2025 Workshop NeurMAD Submission_

### Official Review · Reviewer_jnTH · 2024-12-17
**Review for Submission 1**

**Rating:** 7
**Confidence:** 4

**Review:**

**Paper summary**

SAGE is a self-explainable graph learning framework designed to mitigate distribution shift by “compressing” rather than discarding irrelevant structures, ensuring more stable explanations. By enforcing a lower bound on attention weights and refining parameters through branch optimization, SAGE achieves strong predictive performance and meaningful, interpretable explanations on multiple benchmark datasets.

------

**Originality**

* **Strengths**: SAGE introduces a novel method to control information compression by setting a probabilistic lower bound on edge attention scores, avoiding abrupt distributional shifts caused by pruning. Its branch optimization technique refines model parameters in a straightforward manner without disrupting the main training loop.

* **Weaknesses**: While the concept of controlling information compression is interesting, it is somewhat incremental over other IB-inspired methods. The idea of partial preservation might be seen as a heuristic. The method relies on a hyperparameter r that must be manually chosen and may vary with the dataset.

--------

**Quality**

* **Strengths**: The technical derivation is sound. The model employs a GIN backbone, Gumbel-softmax reparameterization for edge selection, and a KL divergence penalty that encourages attention weights to approximate a predefined distribution. The experiments show performance gains that support the claim.

* **Weaknesses**: The theoretical motivation for why this partial compression (instead of fully dropping edges) leads to better performance could be explored more deeply. The paper lacks rigorous theoretical analysis of how the chosen distribution boundary (r) interacts with distribution shift. The reasoning is intuitive but not extensively justified.

---------

**Significance**

* **Strengths**:  Improving self-explainable GNN performance is a meaningful contribution. The problem of distribution shift is a recognized challenge. Addressing it by partial compression of noisy information could open a new line of thought for balancing model fidelity and interpretability. For practitioners looking for interpretable GNNs with minimal performance loss, this could be valuable.

* **Weaknesses**: Although the results are good, the performance improvements are not always dramatic except on certain datasets (e.g., NCI1). The method’s broad applicability and how it compares to a wide range of other explainability techniques (e.g., recent state-of-the-art methods) is not thoroughly discussed. More extensive baselines or complexity comparisons would add to significance.

-----------

**Questions and suggestions for the authors**

* The performance appears sensitive to the choice of r. Could the authors provide a heuristic or automated method to select r without heavy tuning?
* Could the authors provide more theoretical insights into why partial compression of noisy edges leads to improved performance and reduced distribution shift? For instance, can they characterize how r affects the divergence between compressed and original distributions mathematically?
* The experiments compare with a handful of methods. It would be helpful to see how SAGE compares against a broader range of self-explanatory or post-hoc methods to strengthen claims about general efficacy.
* The training process includes branch optimization and involves Gumbel-softmax sampling. How does the runtime and computational complexity scale with the size of the graph? Are there any memory constraints?

______

**Limitations**

The authors mention distribution shift but do not provide a formal definition or metric beyond similarity in embeddings and a heuristic. While they test multiple datasets, the approach relies on a hyperparameter (r) that must be tuned. The branch optimization step increases computational load and might not be feasible for very large graphs. Another limitation is that while the model provides attention scores as explanations, the granularity and faithfulness of these explanations depend heavily on how well the attention aligns with truly causal substructures. There is also no strong theoretical guarantee provided that the edges identified are indeed the “correct” explanations.

_______

**Ethics**

There are no obvious direct ethical concerns related to the method as it stands. The paper does not deal with sensitive data or produce sensitive content. The approach is a method improvement and not directly involved in human-facing decision-making applications at the evaluation stage. No unethical dataset or methodology usage is apparent. Thus, no ethical issues need to be flagged for special ethics review.

---

### Decision · Program_Chairs · 2024-12-30

**Decision:**

Reject

**Comment:**

This is a good paper. However, it does not fit the scope of this workshop. The SAGE framework has some novelty but is still within the current paradigm of neural networks.